# Ceftazidime-Avibactam Combination Therapy Compared to Ceftazidime-Avibactam Monotherapy for the Treatment of Severe Infections Due to Carbapenem-Resistant Pathogens: A Systematic Review and Network Meta-Analysis

**DOI:** 10.3390/antibiotics9070388

**Published:** 2020-07-07

**Authors:** Marco Fiore, Aniello Alfieri, Sveva Di Franco, Maria Caterina Pace, Vittorio Simeon, Giulia Ingoglia, Andrea Cortegiani

**Affiliations:** 1Department of Women, Child and General and Specialized Surgery, University of Campania Luigi Vanvitelli, 80138 Naples, Italy; anielloalfieri@gmail.com (A.A.); svevadifranco@gmail.com (S.D.F.); caterina.pace@libero.it (M.C.P.); 2Department of Public, Clinical and Preventive Medicine, Medical Statistics Unit, University of Campania Luigi Vanvitelli, 80138 Naples, Italy; vittoriosimeon@gmail.com; 3Department of Surgical, Oncological and Oral Science (Di.Chir.On.S.), Section of Anesthesia, Analgesia, Intensive Care and Emergency, Policlinico Paolo Giaccone, University of Palermo, 90127 Palermo, Italy; giuliaingo92@gmail.com (G.I.); andrea.cortegiani@unipa.it (A.C.)

**Keywords:** carbapenem-resistant Enterobacteriaceae, multidrug resistance, β-lactamase inhibitors, anti-infective agents, bacteremia, ceftazidime-avibactam, sepsis, infection, systematic review, network meta-analysis

## Abstract

Ceftazidime-avibactam (CZA) is a novel beta-lactam beta-lactamase inhibitor combination approved for the treatment of complicated urinary tract infections, complicated intra-abdominal infections, and for hospital-acquired/ventilator-associated pneumonia. The aim of this systematic review (PROSPERO registration number: CRD42019128927) was to evaluate the effectiveness of CZA combination therapy versus CZA monotherapy in the treatment of severe infections. The databases included in the search, until 12 February 2020, were MEDLINE by PubMed, EMBASE, and The Cochrane Central Register of Controlled Trials. We included both randomized controlled trials (RCTs) and non-randomized studies published in peer-reviewed journals and in the English language. The primary outcome was all-cause mortality (longest follow-up) evaluated in patients with the diagnosis of infection with at least one pathogen; secondary outcomes were clinical and microbiological improvement/cure. Thirteen studies were included in the qualitative synthesis: 7 RCTs and 6 retrospective studies All the six retrospective studies identified carbapenamase-producing Enterobacteriaceae (CRE) as the cause of infection and for this reason were included in the network meta-analysis (NMA); the quality of the studies, assessed using the New Castle-Ottawa Scale, was moderate-high. In all the six retrospective studies included in the NMA, CZA was used in large part for off-label indications (mostly blood stream infections: 80–100% of patients included). No difference in mortality rate was observed in patients undergoing CZA combination therapy compared to CZA monotherapy [*n* = 503 patients, direct evidence OR: 0.96, 95% CI: 0.65–1.41].

## 1. Introduction

Ceftazidime-avibactam (CZA) is a combination of the third-generation cephalosporin ceftazidime and a novel non-beta-lactam beta-lactamase inhibitor avibactam. It is a first line therapy for difficult-to-treat infections due to Gram-negative bacilli (GNB) [1]. The prevalence of resistant GNB, including carbapenem-resistant Enterobacteriaceae (CRE) and Pseudomonas aeruginosa, is increasing worldwide. Infections due to CRE are associated with a mortality rate up to 50% [2]. CZA treatment indications are similar in the United States (US) and Europe; the U.S. Food and Drug Administration (FDA) approved CZA in 2015 for the treatment of complicated intra-abdominal infections (cIAI), in combination with metronidazole, and for complicated urinary tract infections (cUTI), including pyelonephritis, in adult patients with limited or no alternative treatment options [3]. The European Medicines Agency (EMA) approved CZA in 2016 for the treatment of adults with cUTIs, cIAIs (in association with metronidazole or an antibacterial agent active against Gram-positive pathogens), and hospital-acquired pneumonia (HAP), including ventilator-associated pneumonia (VAP) [4]. Several studies reported the use of CZA in difficult-to-treat CRE infections, mostly bloodstream infections (BSIs), alone or in combination of other antibiotics [5,6,7,8]. The CZA off-label use in BSIs may be due to the lack of effective therapeutic options. Network meta-analysis (NMA), also known as multiple treatment meta-analysis, has been increasingly used in recent years with the aim to simultaneously compare the effects of multiple treatments on a health-related outcome [9,10]. We performed a systematic review and NMA to compare the effectiveness of CZA mono versus combination therapy with other antibiotics in terms of mortality in patients with CRE infections.

## 2. Methods

The protocol was prospectively registered in PROSPERO (CRD42019128927) on April 16, 2019 [11], after a search of the main electronic registries (Cochrane Database of Systematic Reviews, the JBI Database of Systematic Reviews and Implementation Reports and PROSPERO), to exclude overlap. The present systematic review was conducted according to PRISMA methodology [12]. 

### 2.1. Study Search 

Table 1 shows the review question according to the PICOS format. The databases of the search included MEDLINE via PubMed, EMBASE, and Cochrane Central Register of Controlled Trials (CENTRAL). The full search strategies are reported in the Appendix A. The first search was performed until 4 April 2019; the search was re-run, updating the data collection definitively until 12 February 2020. 

### 2.2. Study Selection

We included both randomized controlled trials (RCTs) and nonrandomized studies (both prospective and retrospective), published in English language and in peer-reviewed journals. No restriction on time of publication was applied.

Figure 1 shows the inclusion–exclusion process according to the preferred reporting items for systematic reviews and meta-analyses (PRISMA) flow diagram. After the searches, duplicates were removed using a citation management software (Endnote VX9. Clarivate Analytics, PA, USA) and a list of all included studies was created. Two authors (SDF and GI) independently performed a screening of retrieved articles based on titles and abstract. The same authors independently evaluated the full texts of the selected articles for final inclusion. The standardized reasons for exclusion were recorded. 

Any disagreements on study eligibility or data extraction were resolved according to a third reviewer’s opinion (AC).

### 2.3. Definition and Outcome

The primary outcome was all-cause mortality evaluated at the longest reported follow-up (if multiple time-points were considered by the authors of the included studies) in the microbiologically evaluable population. The microbiologically evaluable population included all patients who had a diagnosis of infection with at least one pathogen, as confirmed by the laboratory. We did not consider patients with no culture or negative culture. The secondary outcomes were the non-microbiological improvement and the non-clinical improvement.

### 2.4. Data Extraction and Quality Assessment 

Data were extracted from studies included in the review by two reviewers independently (AC, MF) using the Cochrane data collection form for intervention reviews for RCTs and non-RCTs [13]. Two authors assessed the methodological quality of the included studies (AC, MF). The risk of bias of enrolled RCTs was evaluated using the Cochrane Collaboration Revised Assessment Tool (RoB 2) [14]. The quality of nonrandomized studies was assessed using the Newcastle-Ottawa Assessment Scale (NOS) [15].

### 2.5. Data Analysis

We used NMA for synthesizing information and pooled estimates based on the frequentist approach, using the statistical package ‘netmeta’ (version 1.2-0) in R [16]. For a given comparison (A vs B), direct evidence is usually provided by studies that compare these treatments directly. The NMA function provides a back-calculation method to derive indirect estimates from direct pairwise comparisons and combines both direct and indirect evidence across a network of studies into a single effect size. Q and I^2^ statistics were used to quantify heterogeneity among included studies, with I^2^ < 50% indicating no heterogeneity and I^2^ ≥ 50% indicating significant heterogeneity. A random-effects model was conducted in case of significant heterogeneity; otherwise, a fixed-effect model was applied. Q statistic was evaluated for both heterogeneity (within designs) and inconsistency (between designs). Plot of direct evidence proportion for each network estimate was proposed with minimal parallelism and mean path length metrics [17]. Furthermore, a net heat plot was used to evaluate cases of inconsistency between direct and indirect evidence [18].

The main outcome of this NMA and of each single study was mortality. This outcome was pooled as odds ratio (OR) with 95% confidence interval (CI). The ranking probability P-score was used to rank different treatments, where a larger value indicates better performance. This pooled outcome was furthermore analyzed using node splitting method which allows us to control for inconsistency in specific comparisons in our network and graphically represented with a forest plot. All analyses were performed using R software version 3.6.2.

## 3. Results

### 3.1. Study Selection and Characteristics 

We retrieved 2039 articles from the database searches: 1202 on PubMed, 758 on EMBASE, and 79 on CENTRAL. After the removal of 922 duplicates, 1117 articles were identified as potentially relevant and screened as shown in the flow chart (Figure 1). After the screening of titles and abstracts, the full text of 60 articles were evaluated. We further excluded 47 studies for 8 main reasons (Figure 1). Thirteen studies were finally included in the qualitative synthesis (Table 2) and 6 studies in the final meta-analysis (Table 3).

Of the 13 studies included in the qualitative synthesis, of these, 7 were RCT and 6 retrospective cohort studies. The overall risk of bias of the 7 RCT, was low (Appendix A); the quality of the 6 retrospective studies, assessed using the New Castle-Ottawa Scale [15], was moderate-high (Appendix A). Only 2 of the 7 RCT evaluated mortality as outcome [19,20], with a total of 261 patients treated with CZA and 266 not treated with CZA. The microbiological outcome (microbiological non improvement/cure) and the clinical outcome (non clinical improvement/cure) were evaluated in all the 7 RCT. The microbiological outcome, in patients undergoing a pathogen-directed therapy, was evaluated in 2366 patients (1162 patients treated with a CZA-based therapy and 1204 patients treated with other antibiotic treatment). The clinical outcome was evaluated in a total of 2615 patients (1277 patients treated with a CZA-based therapy and 1338 patients treated with other antibiotic treatment).

Three of the 7 RCT enrolled patients with complicated intra-abdominal infections (cIAI) [21,22,23], two enrolled patients with complicated urinary tract infection (cUTI) [24,25], and one enrolled patient with nosocomial pneumonia (NP) [18]. One RCT enrolled patient with both cUTI or cIAI [19]. The 7 RCT enrolled patients infected by bacteria not exclusively belonging to CRE.

Conversely, in all the retrospective studies, the enrolled patients were infected by CRE only; in 4 retrospective studies, the authors enrolled patients with infections caused by all CRE [6,8,26,27] and in 2 studies, only Klebsiella pneumoniae carbapenemase (KPC)-producing [5,7], Klebsiella pneumoniae belongs to the tribe Klebsiellae, a member of the Enterobacteriaceae family [28]. In the studies included in the NMA, the authors have adopted a phenotypic definition of resistance to carbapenems (i.e., based on the antibiotic susceptibility pattern). The focus of infection in the 83% (419/503) of patients was bacteremia; only in 3 of these 6 retrospective studies bacteremia was aggregated with other focus of infection: In the study by King et al. [26], two fifths of the patients (23/60) had bacteremia, in the study by Sousa et al. [8], almost half of the patients (26/57) have bacteremia. In the study by Alraddadi et al. [27], more than a half of patients (22/38) have bacteremia. 

For the quantitative evaluation, we decided to meta-analyze the studies in which the infection was caused by pathogens resistant to carbapenems.

### 3.2. NMA Heterogeneity and Inconsistency Evaluation

There is no statistical heterogeneity among included studies, with I^2^ 0% (95% CI; 0–17.9%) and thereby a fixed-effect model was applied. The Q statistic was 0.62 (*p* = 0.89) for the within design and 1.48 (*p* = 0.69) for the between design indicating again no heterogeneity and consistency of model used.

As showed in Figure 2, more than 75% of each comparison is represented by direct proportion. Minimal parallelism is always higher than 1 and mean path length lower than 2, showing good metrics as reported by Konig and colleagues [17]. The net heat plot confirms the consistency of data and studies in this NMA [29] (Figure 3).

### 3.3. Mortality

We planned to analyze the longest follow-up, if the mortality was reported at different time-points; in 5 of 6 retrospective studies, included in the NMA, the longest follow-up was 30-day-mortality (Table 2). 

The NMA, including 503 patients, did not show significant differences in CZA-combination therapy compared to CZA-monotherapy [OR: 0.96, 95% CI: 0.65–1.41]. A significant difference was observed in both CZA-monotherapy vs. best available antibiotic therapy (BAT) [OR: 0.68, 95% CI: 0.48–0.98] and CZA-combination vs. BAT [OR: 0.66, 95% CI: 0.48–0.89], the forest plot is shown in Figure 4. As a whole, the ranking probability P-score showed a larger value for CZA-combination therapy (0.7917) and CZA-monotherapy (0.6965) with the lowest rank score for the BAT (0.0118). 

Non-clinical improvement was evaluated in only 4 studies [6,8,26,27], including 186 patients and microbiological failure was evaluated in only 4 studies [5,8,26,27], including 363 patients (Table 2). Therefore, due to the small number of patients, we did not proceed to quantitative synthesis for the secondary outcomes.

## 4. Discussion

The main findings of this study were that there were no significant differences in mortality in the treatment of CRE infections with CZA-combination therapy compared to CZA-monotherapy, based on available evidence.

Transmissible carbapenem-resistance in Enterobacteriaceae, among which the KPCs are the most notorious, has been reported since twenty years, but only recently it has been expressed as a public health problem with outbreaks reported worldwide [31,32,33,34]; CRE, due to the difficulty of effective treatment and the very high attributable mortality, are also known as “nightmare bacteria” [2]. Infections caused by these bacteria are associated with a mortality rate exceeding 50%. Since the burden of this problem is dramatic, the introduction of new antibiotics or the alternative use of existing antibiotics, as well as in our study exploring the off-label use of CZA in BSIs, is indispensable. 

The Infectious Disease Society of America recommends the development of new antibiotic options through pathogen-directed studies, in which patients with multiple disease types are enrolled, rather than a single type of infection [35].

For these premises, we have meta-analyzed cumulates results from pathogen-directed studies, enrolling patients with different types of infection; for more than four-fifths of the patients, the infection focus was a BSI. Therefore, although the number of included patients is relatively low, the review cohort is homogeneous. We decided to explore as primary outcome the all-cause mortality because it is highly objective, accurate, and simple to measure [36], especially in case of low-quality evidence. 

No difference in mortality rate was observed in patients undergoing CZA combination therapy compared to those who received CZA monotherapy for the treatment of CRE infections. This finding may be useful for optimizing the antibiotic treatment, with the potential to reduce the use of combination treatments [37].

Our results should be considered in light of some limitations. The outcome of mortality was not assessed in all studies at the same time point, although in the vast majority of studies (83%: 5/6), it was assessed at 30 days (Table 3). We used raw data from observational studies, and this approach is prone to the effect of potential confounders [38]. Our results cannot be extrapolated to infections other than due to CRE and should be carefully considered when treating patients with high severity of diseases. Moreover, local epidemiology should be considered when deciding to use CZA mono- or combination therapy. Due to the small number of studies and patients, we were not able to explore the secondary outcomes. Furthermore, we did not consider the side effect of antibiotics among the outcomes [11]. A limitation of our study is that a phenotypic diagnosis of resistance to carbapenems was adopted; there are many different mechanisms (i.e., genotypes) that can result in carbapenem resistance, while phenotypic tests are easy and cost-effective to perform, molecular diagnostic techniques can tailor treatment guidelines to optimize patient’s management [39].

Our results are in line with an unregistered systematic review and meta-analysis without network comparison but with a precedent search date [40].

In conclusion, in this systematic review and NMA CZA monotherapy was as effective as CZA combinations in reducing all-cause mortality in patients with infections by CRE (mostly KPC) but the quality of the available evidence and the overall number of patients from included studies was low. 

Further clinical trials should evaluate the effectiveness and safety of different CZA combinations, especially in other infections and clinical settings. Moreover, further evidence is likely to change the outcome estimates.

## Figures and Tables

**Figure 1 antibiotics-09-00388-f001:**
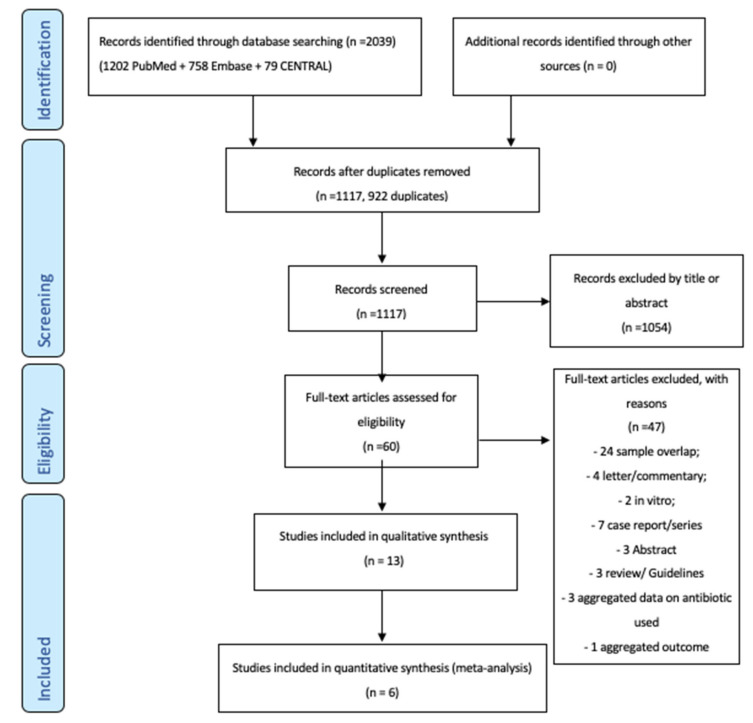
The flow-chart of the study selection.

**Figure 2 antibiotics-09-00388-f002:**
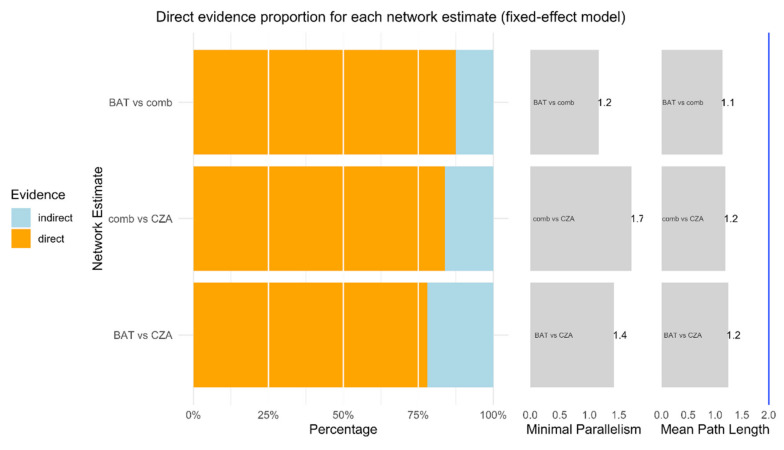
This function plots relevant measures quantifying the percentage of direct and indirect evidence proportion, the aggregated minimal parallelism, and a mean path length of a frequentist network meta-analysis model [30].

**Figure 3 antibiotics-09-00388-f003:**
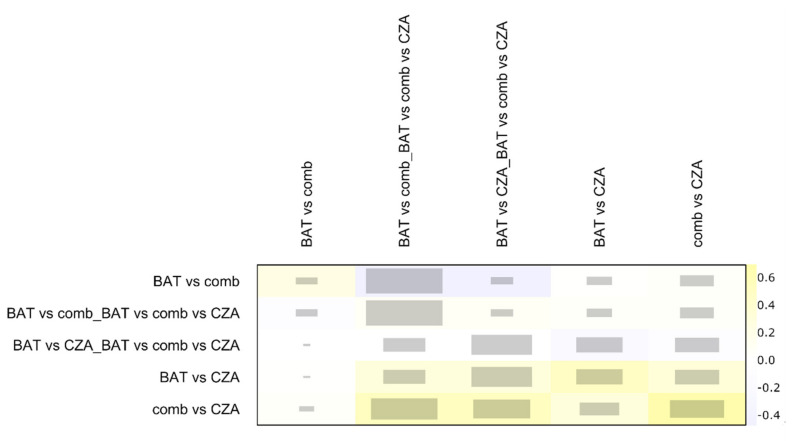
The graph is a quadratic matrix in which each element in a row is compared to all other elements in the columns. Treatment comparison with only one kind of evidence (i.e., indirect or indirect evidence) are omitted in this plot. The grey boxes for each comparison of designs signify how important a treatment comparison is for the estimation of another treatment comparison. The bigger the box, the more important a comparison is. The colored backgrounds, ranging from blue to red, signify the inconsistency of the comparison in a row attributable to the design in a column. Red is highly inconsistent; blue is a lower value of inconsistency. BAT: Best antibiotic therapy; CZA: Ceftazidime-avibactam monotherapy; comb: Ceftazidime-avibactam combination therapy.

**Figure 4 antibiotics-09-00388-f004:**
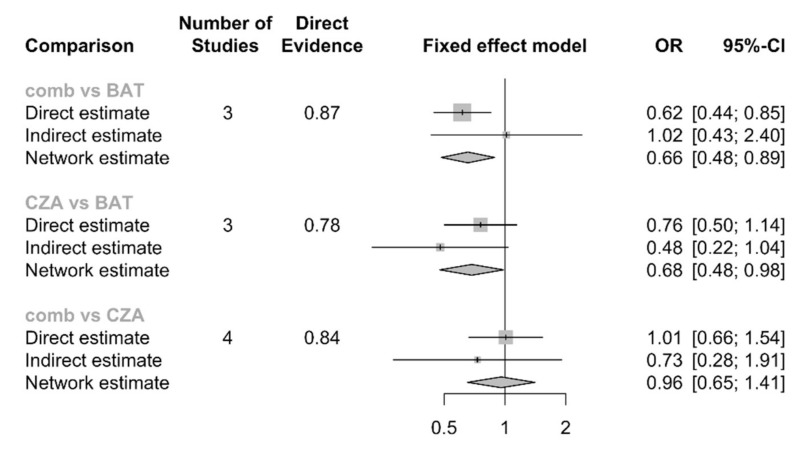
Number of studies describe how many studies were analyzed for that comparison, with proportion of direct evidence. Direct estimate represents OR and 95%CI directly calculated from real comparison (for example CZA vs. BAT); Indirect estimation is a back-calculation based on other comparison (CZA vs. BAT estimated through CZA vs. comb and BAT vs. comb); Network estimation is a back-calculation based on direct and indirect evidence. BAT: Best antibiotic therapy; CZA: Ceftazidime-avibactam monotherapy; comb: Ceftazidime-avibactam combination therapy.

**Table 1 antibiotics-09-00388-t001:** PICOS method for selecting clinical studies in the systematic reviews.

Participants	Intervention	Comparison	Outcomes	Study Design
Adult patients in any setting with confirmed bacterial infection	Ceftazidime-avibactam in association with another antibiotic/s	Ceftazidime-avibactam alone	Primary outcomes: all-cause mortality Secondary outcomes: (a) not clinical improvement, (b) not microbiological improvement	Randomized controlled trials and observational studies (including cohort andcase–control studies)

**Table 2 antibiotics-09-00388-t002:** Summary of the studies included in the qualitative synthesis.

Author (Published Year) [ref.]	Journal	Study Design	Time Spam	Pathogen	Septic Focus	Evaluation Time Points		
						Mortality	Clinical	Microbiological
Sousa (2018) [8]	J Antimicrob Chemother	RC	04/16–12/17	CRE (KPC: 95%)	Mix	30-days	7-days	7-days
King (2017) [26]	Antimicrob Agents Chemother	RC	03/05–04/16	CRE (KPC: 83%)	Mix	In Hospital	end of therapy	end of therapy
Alraddadi (2019) [27]	BMC Infectious Diseases	RC	01/17–08/18	CRE (KPC: 79%)	Mix	30-days	30-days	30-days
Caston (2017) [6]	Int J Infect Dis	RC	06/12–03/16	CRE (KPC: 80%)	Bacteremia	30-days	14-days	-
Shields (2017) [7]	Antimicrob Agents Chemother	RC	01/09–02/17	CRE (KPC: 100%)	Bacteremia	30-days	-	-
Tumbarello (2019) [5]	Clin Infect Dis	RC	04/16–12/17	CRE(KPC: 100%)	Bacteremia	30-days	-	In Hospital*
Carmeli (2016) [19]	Lancet Infect Dis	RCT	01/13–08/14	MIX	cIAI / cUTI	28-days	7–10 days	21/25-days
Lucasti (2013) [21]	J Antimicrob Chemother	RCT	03/09–12/09	MIX	cIAI	-	14-days	14-days
Qin (2017) [22]	Int J Antimicrob Agents	RCT	01/13–03/15	MIX	cIAI	-	28–35 days	-
Mazuski (2016) [23]	Clin Infect Dis	RCT	03/12–04/13	MIX	cIAI	-	28–35 days	28–35 days
Vazquez (2012) [24]	Curr Med Res Opin	RCT	11/08–06/10	MIX	cUTI	-	5–9 days	5–9 days
Wagenlehner (2016) [25]	Clin Infect Dis	RCT	10/10–08/14	MIX	cUTI	-	21–25 days	21–25 days
Torres (2018) [20]	Lancet Infect Dis	RCT	04/13–02/15	MIX	NP	21–25 days	21–25 days	21–25 days

CRE: carbapenem-resistant Enterobacteriaceae; KPC: Klebsiella pneumoniae carbapenemasi; RC: Retrospective cohort study; RCT: Randomized controlled trial; Mix: Mixed bacterial flora; MIX: Aggregated foci of infection.

**Table 3 antibiotics-09-00388-t003:** Summary of the characteristics of enrolled studies in the meta-analysis.

Author (Published Year) [ref.]	N° of patients Enrolled	N° of Bacteremia (%)	N° of Patients Treated with CZA Alone	N° of Patients Treated with CZA Association	N° of Patients Treated with BAT	BAT	CZA-Associated Antibiotic	Medical Ward
Sousa (2018) [8]	57	26 (46)	46	11	X	X	#	NS
King (2017) [26]	60	23 (38)	33	27	X	X	^	Mix ICU (59%)
Alraddadi (2019) [27]	38	22 (58)	10	X	28	§	X	NS
Caston (2017) [6]	31	31 (100)	X	8	23	^^	^^	Mix ICU (10%)
Shields (2017) [7]	109	109 (100)	8	5	96	++	<>	Mix ICU (50%)
Tumbarello (2019) [5]	208	208 (100)	22	82	104	§§	##	Mix ICU (33.3%)

# colistin iv 5, inhaled colistin 2, tigecycline 2, amikacin 1, imipenem 1. ^ aminoglycosides 40%, polymyxin 26%, tigecycline 22%. § colistin (21, 75%), carbapenem (21, 75%), tigecycline (9, 32.1%), aminoglycoside (8, 28.6%), quinolone (4, 14.3%), trimethoprim/sulfamethoxazole (1, 3.6%) and aztreonam (1, 3.6%). ^^association of: aminoglycoside (*n* = 7, 87.5%), carbapenems (n = 3, 37.5%), fosfomycin (*n* = 2, 25%), tigecycline (*n* = 2, 25%), colistin (*n* = 2, 25%). ++ carbapenem plus colistin (*n* = 30); carbapenem plus aminoglycoside (*n* = 25); monotherapy consisted of an aminoglycoside (*n* = 11), carbapenem (*n* = 8), colistin (*n* = 4), tigecycline (*n* = 4), ciprofloxacin (*n* = 2); Combination regimens were colistin plus tigecycline (*n* = 3), aminoglycoside plus tigecycline (*n* = 2), and 1 each of aminoglycoside plus cefepime, aminoglycoside plus colistin plus tigecycline, colistin plus aztreonam, colistin plus cefepime, colistin plus ciprofloxacin, carbapenem plus doxycycline, and carbapenem plus tigecycline. <> gentamicin. §§ Double carbapenem (*n* = 29, 27.9%); Gentamycin (*n* = 14, 13.5%); Fosfomycin plus amikacin (*n* = 13, 12.5%); Fosfomycin plus gentamycin (*n* = 11, 10.5%); Gentamycin plus meropenem (*n* = 11, 10.5%); Colistin plus fosfomycin (*n* = 10, 9.6%); Colistin (*n* = 9, 8.6%); Others (*n* = 7, 6.7%). ## Noncarbapenem drugs (*n* = 63, 76.8%); carbapenem (*n* = 19, 23.17%). Carbapenem-containing regimens included imipenem OR meropenem with or without ertapenem. BAT: Best antibiotic therapy; CZA: Ceftazidime-avibactam; ICU: Intensive care unit; NS: Not specified; Mix: Different type of medical wards.

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
