# Peer review of "Ceftazidime-Avibactam Combination Therapy Compared to Ceftazidime-Avibactam Monotherapy for the Treatment of Severe Infections Due to Carbapenem-Resistant Pathogens: A Systematic Review and Network Meta-Analysis"

_antibiotics, 2020, doi:10.3390/antibiotics9070388_

Round 1
Reviewer 1 Report
The authors systematically reviewed CZA combination therapy with CZA monotherapy in the treatment of severe infections. The introduction is easy to understand and provides an aim for orientation. The methods / results are sufficiently described. The discussion embedded the results into a broader context. I found it interesting to read. The only issue I really missed was the fact that the analysis reports the mortality rate but at the same time it could also have reported moderate to severe adverse clinical effects which might be important to know.
Author Response
Reviewer 1 Comments
Comment: The authors systematically reviewed CZA combination therapy with CZA monotherapy in the treatment of severe infections. The introduction is easy to understand and provides an aim for orientation. The methods / results are sufficiently described. The discussion embedded the results into a broader context. I found it interesting to read. The only issue I really missed was the fact that the analysis reports the mortality rate but at the same time it could also have reported moderate to severe adverse clinical effects which might be important to know.
Reply 1: We thank reviewer #1 for the comment, The protocol was prospectively registered in PROSPERO (CRD42019128927) on April 16, 2019 and we did not consider side effects of the relevant antibiotics among the review outcomes. We selected all-cause mortality as primary outcome because it is highly objective, accurate, and simple to measure, especially in case of low-quality evidence (lines 309-311).
Following your comment, we stated it in the discussion among the limitations of our systematic review (lines 323-324). Quote: “we did not consider the side effect of antibiotics among the outcomes”.
Reviewer 2 Report
My comments/suggestions are as follows:
General comments:
- A brief introduction about network meta analysis should be included in the introduction.
- Entire 'Discussion' section is riddled with grammatical and sentence construction errors. This section should be completely rewritten.
- Getting the manuscript proofread by a native English speaker or an editing professional is highly recommended.
Specific comments:
- Line 89-91: Sentence meaning not clear. Consider splitting up the sentence into two.
- Line 97-99: Do the authors mean Figure 1 as the PRISMA flow diagram? If so, please mention it.
- Line 102-103: 'if were reported........analyze the longest one'. Consider reconstructing this sentence.
- Line 103: The authors mention 'microbiologically evaluable' population. Does this mean the microbiologically none valuable patients (in other words culture negative) were excluded from the study. But table one does list in the 'Participants' section as 'confirmed or suspected' bacterial infection. Please be explicit about this in line 103, section 'Definition and outcome'.
- Line 138-139: '1054 studies were excluded'. Based on what?
- Line 140: '8 main reasons'. Please list the reasons in the main text as well, in addition to the figure. Alternately, point readers to Figure 1 after this sentence.
- Line 141-142: This statement should come after the paragraph line 143-146. Right now, the inclusion criteria comes first and the explanation next. The correct flow of text should be the other way around.
- Line 157-158: 'In all RCT.......same resistance profile'. Difficult to follow. Please reconstruct this sentence.
- Line 247-248: No meaning is coming out of this line. The authors first mention the 'P-score for CZA combination therapy to be higher (0.7917) than CZA alone (0.6965)', and then say 'on the contrary was low for the combination therapy (0.0118)'. I could not understand what the authors are trying to convey.
- Figure 2: Please provide a more detailed legend explaining the figure.
- Table 3 and figure 3: Please expand 'BAT' as best antibiotic therapy in the table and figure legends.
- Line 271: 'in 5 of 6 included studies it 30-day-mortality' should be 'in 5 of 6 included studies it was 30-day-mortality'.
- Line 287-289: Meaning not clear. Please rewrite.
- Line 290-297: Difficult to understand. Sentence constructions and flow of text needs to be thoroughly revised.
Author Response
Reviewer 2 Comments
Comment: A brief introduction about network meta analysis should be included in the introduction.
Reply 1: We included a brief introduction about network meta-analysis as suggested (lines 61-63)
Comment: Entire 'Discussion' section is riddled with grammatical and sentence construction errors. This section should be completely rewritten.
Reply 2: We eliminated grammatical errors in our discussion
Comment: Getting the manuscript proofread by a native English speaker or an editing professional is highly recommended.
Reply 3: The manuscript was revised by a native English speaker.
Comment: Line 89-91: Sentence meaning not clear. Consider splitting up the sentence into two.
Reply 4: We spitted the sentence in two (lines 97-99). Quote: “Two authors (SDF and GI) independently performed a screening of retrieved articles based on titles and abstract. The same authors independently evaluated the full texts of the selected articles for final inclusion. The standardized reasons for exclusion were recorded”
Comment: Line 97-99: Do the authors mean Figure 1 as the PRISMA flow diagram? If so, please mention it.
Reply 5: We mentioned the Figure 1 in the text (lines 94-95).
Comment: Line 102-103: 'if were reported........analyze the longest one'. Consider reconstructing this sentence.
Reply 6: We rephrased the sentence (lines 104-106). Quote: “The primary outcome was all-cause mortality at the longest reported follow-up (if multiple time-points were considered by the authors of the included studies) in the microbiologically evaluable population”
Comment: Line 103: The authors mention 'microbiologically evaluable' population. Does this mean the microbiologically none valuable patients (in other words culture negative) were excluded from the study. But table one does list in the 'Participants' section as 'confirmed or suspected' bacterial infection. Please be explicit about this in line 103, section 'Definition and outcome'.
Response 7: We thank you the Reviewer for giving us the chance to fix this error. we deleted in table one “or suspected' bacterial infection (line 86) and explicated the microbiologically evaluable population definition (line 106-108, ex line 103)
- Line 138-139: '1054 studies were excluded'. Based on what?
Response 8: We added the words “by title or abstract” in the figure (line 146) and “After the screening of titles and abstracts” in the text (lines 142-143)
- Line 140: '8 main reasons'. Please list the reasons in the main text as well, in addition to the figure. Alternately, point readers to Figure 1 after this sentence.
Response 9: We decided to point readers to Figure 1 after the sentence (line 144)
- Line 141-142: This statement should come after the paragraph line 143-146. Right now, the inclusion criteria comes first and the explanation next. The correct flow of text should be the other way around.
Response 10: We moved the statement as suggested (lines 144-145)
- Line 157-158: 'In all RCT.......same resistance profile'. Difficult to follow. Please reconstruct this sentence.
Response 11: We reconstructed the sentence as suggested The 7 RCT enrolled patients infected by bacteria not exclusively belonging to enterobacteriaceae resistant to carbapenems (CRE). (lines 162-163)
- Line 247-248: No meaning is coming out of this line. The authors first mention the 'P-score for CZA combination therapy to be higher (0.7917) than CZA alone (0.6965)', and then say 'on the contrary was low for the combination therapy (0.0118)'. I could not understand what the authors are trying to convey.
Response 12: The description of the P-score is in the methods, it represents a ranking of the treatments (e.g. first, second and third place), we moved and rephrased the sentence to improve the clarity of the text (lines 277-278)
- Figure 2: Please provide a more detailed legend explaining the figure.
Response 13: We provided a more comprehensible legend and added a reference for Details
- Table 3 and figure 3: Please expand 'BAT' as best antibiotic therapy in the table and figure legends.
Response 14: We expanded BAT in the legends as requested (lines 238, 266, 291-292)
- Line 271: 'in 5 of 6 included studies it 30-day-mortality' should be 'in 5 of 6 included studies it was 30-day-mortality'.
Response 15: Many thanks, we corrected as suggested (line 271)
- Line 287-289: Meaning not clear. Please rewrite.
Response 16: We rephrased as suggested (lines 295-297); Quote “The main findings of this study were that there were no significant differences in mortality in the treatment of CRE infections with CZA-combination therapy compared to CZA-monotherapy, based on available evidence”.
- Line 290-297: Difficult to understand. Sentence constructions and flow of text needs to be thoroughly revised.
Response 17: We rephrased as suggested (lines 298-304); Quote “Transmissible carbapenem-resistance in Enterobacteriaceae, among which the KPCs are the most notorious, has been reported since twenty years, but only recently it has expressed as a public health problem with outbreaks reported worldwide; CRE, due to the difficulty of effective treatment and the very high attributable mortality, are also known as “nightmare bacteria”. Infections caused by these bacteria are associated with a mortality rate exceeding 50%. Since the burden of this problem is dramatic, the introduction of new antibiotics or the alternative use of existing antibiotics, as well as in our study exploring the off-label use of CZA in BSIs, is indispensable”

Reviewer 3 Report
- Brief summary
The authors report an original manuscript that main to evaluate the effectiveness of Ceftazidime-avibactam (CZA) combination therapy versus CZA monotherapy in the treatment of severe infections. The main conclusion of the manuscript is that was not found significant mortality differences in the treatment of CRE infections with CZA-combination therapy compared to CZA-monotherapy.
- Broad comments
- It is a very relevant topic, considering that increase the knowledge about effectiveness of new antibiotics and/or β-lactam/ β-lactamase inhibitors combinations, are urgently needed into the clinical practice, especially to apply on severe infections by carbapenems-resistant Enterobacteriaceae that has a mortality rate up to 50%.
- It is a well written manuscript and the sections are well organized and include the relevant topics nevertheless some recommendations should be considered by the authors.
- The Abstract is concise and adequately summarizes the article content. However, the sentence (line 30-32) is unclear and should be revised: “The six RS, homogeneous regarding the causative pathogen (carbapenamase-producing Enterobacteriaceae: CRE) were included in the network meta-analysis (NMA):”
- The authors have decided to meta-analyze the studies in which the infection was caused by pathogens resistant to carbapenems (lines 166-167). Previously, have described that “(…) in all the RC the bacteria which were the cause of the infection were resistant to carbapenems; in 4 studies were enterobacteriaceae resistant to carbapenems (CRE) [8,25] [26] [6] and in 2 studies were Klebsiella pneumoniae carbapenemase (KPC)-producing [5,7].” (lines 158-161).
However, the production of KPC enzyme can be produced by different pathogens, not only Enterobacteriaceae resistant to carbapenems. Additionally, the pathogens should be identified (were Escherichia coli? Klebsiella pneumoniae? Enterobacter?). Moreover, the KPC-type enzymes can be KPC-3, KPC-2 or others.
It is important to note that also in Table 2 (line 215) the authors include the Pathogen and identify a “Mix” – that is not described at the manuscript and identify “CRE1 (KPC2 X%)” but once again there is not description of which CRE, the type of KPC and the correlation of CRE-KPC. The legend of 1 and 2 is inexistent.
This is a major consideration that should be reviewed by the authors in order to promote the accuracy of the manuscript for the readers but also because the authors conclude that “CZA monotherapy may be as effective as CZA combinations in reducing mortality in infections due to CRE, including KPC, but the quality of the available evidence and the overall number of studied patients is low” (lines 322-324).
- Future studies should be included at discussion.
- The references should be extensively reviewed. For example:
6.1. The reference 2 (line 351) is not correctly presented: (CDC), C.f.D.C.a.P. Clinicians: Information about CRE. 2019
6.2. The reference 3 and 4 are not completed: “(accessed on)”.
- It is suggested a English review of the manuscript.
- Other minor suggestions to the authors:
- The word “antibiotic/s” (line 59) should be corrected.
- enterobacteriaceae should be corrected to Enterobacteriaceae (line 159).
- What is “1 Ca” ? (line 216)
- “aminoglycoside plu tigecycline” should be corrected (line 230)
- [[36]] (line 315) should be corrected.
Author Response
Reviewer 3 Comments
- Brief summary
The authors report an original manuscript that main to evaluate the effectiveness of Ceftazidime-avibactam (CZA) combination therapy versus CZA monotherapy in the treatment of severe infections. The main conclusion of the manuscript is that was not found significant mortality differences in the treatment of CRE infections with CZA-combination therapy compared to CZA-monotherapy.
- Broad comments
- It is a very relevant topic, considering that increase the knowledge about effectiveness of new antibiotics and/or β-lactam/ β-lactamase inhibitors combinations, are urgently needed into the clinical practice, especially to apply on severe infections by carbapenems-resistant Enterobacteriaceae that has a mortality rate up to 50%.
Response 1:
Thank you for your comment.
- It is a well written manuscript and the sections are well organized and include the relevant topics nevertheless some recommendations should be considered by the authors.
Response 2: Thanks for the helpful suggestions.
- The Abstract is concise and adequately summarizes the article content. However, the sentence (line 30-32) is unclear and should be revised: “The six RS, homogeneous regarding the causative pathogen (carbapenamase-producing Enterobacteriaceae: CRE) were included in the network meta-analysis (NMA):”
Response 3: We rephrased the sentence in “All the six retrospective studies identified carbapenamase-producing Enterobacteriaceae (CRE) as the cause of infection and for this reason were included in the network meta-analysis (NMA)” (lines 31-33)
- The authors have decided to meta-analyze the studies in which the infection was caused by pathogens resistant to carbapenems (lines 166-167). Previously, have described that “(…) in all the RC the bacteria which were the cause of the infection were resistant to carbapenems; in 4 studies were enterobacteriaceae resistant to carbapenems (CRE) [8,25] [26] [6] and in 2 studies were Klebsiella pneumoniae carbapenemase (KPC)-producing [5,7].” (lines 158-161).
Response 4: We rephrased the sentence “in 4 retrospective studies the authors enrolled patients with infections caused by all enterobacteriaceae resistant to carbapenems (CRE) and in 2 studies only Klebsiella pneumoniae carbapenemase (KPC)-producing, Klebsiella pneumoniae belongs to the tribe Klebsiellae, a member of the enterobacteriaceae family” (lines 164-167)
However, the production of KPC enzyme can be produced by different pathogens, not only Enterobacteriaceae resistant to carbapenems. Additionally, the pathogens should be identified (were Escherichia coli? Klebsiella pneumoniae? Enterobacter?). Moreover, the KPC-type enzymes can be KPC-3, KPC-2 or others.
Response 5: We specified that “In the studies included in the NMA the authors have adopted a phenotypic definition of resistance to carbapenems (i.e., based on the antibiotic susceptibility pattern)” (lines 167-169). In the discussion we stated that “A limitation of our study is that a phenotypic diagnosis of resistance to carbapenems was adopted; there are many different mechanisms (i.e., genotypes) that can result in carbapenem resistance, while phenotypic tests are easy and cost-effective to perform, molecular diagnostic techniques can tailor treatment guidelines to optimize patient's management” (lines 326-329)
It is important to note that also in Table 2 (line 215) the authors include the Pathogen and identify a “Mix” – that is not described at the manuscript and identify “CRE1 (KPC2 X%)” but once again there is not description of which CRE, the type of KPC and the correlation of CRE-KPC. The legend of 1 and 2 is inexistent.
Response 6: We apologize for the typo, we rearranged the legend of the Table 2 (lines 213-214)
This is a major consideration that should be reviewed by the authors in order to promote the accuracy of the manuscript for the readers but also because the authors conclude that “CZA monotherapy may be as effective as CZA combinations in reducing mortality in infections due to CRE, including KPC, but the quality of the available evidence and the overall number of studied patients is low” (lines 322-324).
Response 7: Thank you for this important comment. To improve the accuracy of our conclusions, we modified the sentence as follows: Line 332-334 “In conclusion, in this systematic review and NMA CZA monotherapy was as effective as CZA combinations in reducing all-cause mortality in patients with infections by CRE (mostly KPC) but the quality of the available evidence and the overall number of patients from included studies was low”
- Future studies should be included at discussion.
Response 8: We included at discussion the sentence “Further clinical trials should evaluate the effectiveness and safety of different CZA combinations, especially in other infections and clinical settings. Moreover, further evidence is likely to change the outcome estimates.” (lines 335-337)
- The references should be extensively reviewed. For example:
6.1. The reference 2 (line 351) is not correctly presented: (CDC), C.f.D.C.a.P. Clinicians: Information about CRE. 2019
Response 9: We corrected as suggested (line 467)
6.2. The reference 3 and 4 are not completed: “(accessed on)”.
Response 10: We added “July 3, 2020” (lines 366, 368)
- It is suggested a English review of the manuscript.
Response 11: The manuscript was revised by a native English speaker
- Other minor suggestions to the authors:
- The word “antibiotic/s” (line 59) should be corrected.
Response 12: We rephrased the sentence “We performed a systematic review and NMA to compare the effectiveness of CZA mono- versus combination therapy with other antibiotics in terms of mortality in patients with CRE infections” (lines 63-65)
- enterobacteriaceae should be corrected to Enterobacteriaceae (line 159).
Response 13: We corrected in CRE (line 164)
- What is “1 Ca” ? (line 216)
Response 14: We apologize, it’s a typo.
- “aminoglycoside plu tigecycline” should be corrected (line 230)
Response 15: we corrected in “aminoglycoside plus tigecycline” (line 228)
- [[36]] (line 315) should be corrected.
Response 16: We corrected it (line 321)

Round 2
Reviewer 3 Report
The authors have significantly improved the manuscript.